# Insulin-like peptides and the mTOR-TFEB pathway protect *Caenorhabditis elegans* hermaphrodites from mating-induced death

Cheng Shi[1,2], Lauren N Booth[3], Coleen T Murphy[1,2]*

[1]Department of Molecular Biology, Princeton University, Princeton, United States; [2]LSI Genomics, Princeton University, Princeton, United States; [3]Department of Genetics, Stanford University, Stanford, United States

**Abstract** Lifespan is shortened by mating, but these deleterious effects must be delayed long enough for successful reproduction. Susceptibility to brief mating-induced death is caused by the loss of protection upon self-sperm depletion. Self-sperm maintains the expression of a DAF-2 insulin-like antagonist, INS-37, which promotes the nuclear localization of intestinal HLH-30/TFEB, a key pro-longevity regulator. Mating induces the agonist INS-8, promoting HLH-30 nuclear exit and subsequent death. In opposition to the protective role of HLH-30 and DAF-16/FOXO, TOR/LET-363 and the IIS-regulated Zn-finger transcription factor PQM-1 promote seminal-fluid-induced killing. Self-sperm maintenance of nuclear HLH-30/TFEB allows hermaphrodites to resist mating-induced death until self-sperm are exhausted, increasing the chances that mothers will survive through reproduction. Mothers combat males' hijacking of their IIS pathway by expressing an insulin antagonist that keeps her healthy through the activity of pro-longevity factors, as long as she has her own sperm to utilize.

DOI: https://doi.org/10.7554/eLife.46413.001

*For correspondence:
ctmurphy@princeton.edu

**Competing interests:** The authors declare that no competing interests exist.

## Introduction

The battle between the sexes is played out across the animal kingdom: males compete with one another to pass on their own genomes, often at the expense of female health, and females, in turn, must devise strategies to reduce the deleterious effects of mating in order to successfully complete reproduction. Sexual antagonism is evident in *Caenorhabditis elegans*, where signals from males kill hermaphrodites through three distinct mechanisms: germline activation (*Shi and Murphy, 2014*), seminal fluid transfer (*Shi and Murphy, 2014*), and, at high doses, male pheromone toxicity (*Maures et al., 2014*; *Shi et al., 2017*). *Caenorhabditis elegans* are hermaphrodites and thus do not require males to reproduce; in fact, males are normally rare in nature (~1/1000). Brief bursts of mating with males can be beneficial for the species due to increased outcrossing (*Kimble, 1988*), but males are quickly eliminated by dispersal (*Wegewitz et al., 2008*; *Anderson et al., 2010*) and by their own pheromones (*Shi et al., 2017*).

While germline activation is necessary for reproduction and the resulting lifespan decrease is shared both across sexes and species, seminal fluid (SF) killing appears to be used specifically for the purpose of 'sperm competition' (that is, competition between males) within the species: seminal fluid transfer induces death of mothers just after they have produced all of that male's progeny, eliminating the possibility of the mother re-mating with a different male (*Shi and Murphy, 2014*). We noted previously that mating-induced death occurs only after *C. elegans* hermaphrodites produce all their progeny (*Shi and Murphy, 2014*); this delay is necessary for successful reproduction, but the

underlying mechanism allowing this balance was previously unknown. In an accompanying manuscript, Booth et al. report that self-sperm are protective against short-term mating-induced death specifically in hermaphroditic mothers (*Booth et al., 2019*). Here, we describe the molecular mechanism that hermaphrodites use to prevent early SF-induced death until self-sperm are depleted. This pathway utilizes competing insulin-like peptide agonists and antagonists of the DAF-2/Insulin receptor as well as mTOR signaling to regulate a set of protective (DAF-16/FOXO and HLH-30/TFEB) and detrimental (PQM-1) transcription factors, which in turn determine susceptibility to seminal-fluid-induced death.

## Results

### Self-sperm protects young hermaphrodites from male seminal-fluid-induced death

The presence of self-sperm in young hermaphrodites protects them from brief mating-induced death (*Booth et al., 2019*; *Figure 1A,B*). Unlike older hermaphrodites, young hermaphrodites that mate during a brief interaction with males are protected from this mating-induced death (*Booth et al., 2019*; *Figure 1A,B*). By contrast, young feminized (i.e. lacking self-sperm; *fog-2(q71), fem-1(hc17)*) worms exhibit significant acceleration of death after brief mating with males (*Booth et al., 2019*; *Figure 1C*, *Figure 1—figure supplement 1A*), suggesting that self-sperm are protective.

Previously we found that mating induces hermaphrodite shrinking through male sperm activation of the hermaphrodite's germline, and that separately, transfer of male seminal fluid (SF) causes DAF-16/FOXO cytoplasmic localization and inactivation, subsequently leading to death of the mother (*Shi and Murphy, 2014*). To determine whether brief mating acts through the male sperm/germline pathway or through the seminal fluid pathway, we mated *fer-6* males, which are spermless but still contain seminal fluid (*Figure 1D,E*; *Argon and Ward, 1980*; *Gems and Riddle, 1996*), or *fog-2* males, which contain male sperm and thus essentially are like wild type (*Schedl and Kimble, 1988*), with *fog-2* (self-spermless) females; both sperm-containing and spermless males induce death in young and old *fog-2* mothers at equal rates, suggesting that the death-inducing factor is not sperm. The dispensability of male sperm suggests that brief mating-induced death acts through the seminal fluid pathway.

Because high doses of male pheromones can also kill hermaphrodites (*Maures et al., 2014*; *Shi et al., 2017*), we tested whether pheromone-less *daf-22* (*Ludewig and Schroeder, 2013*) males can also induce killing after brief mating. However, *daf-22* males induce the same brief mating lifespan effect as wild-type males. Similarly, 2 hr treatment on high-dose male pheromone-conditioned plates ('MCP'; plates pre-treated with 30 males, then removed prior to experiment) does not shorten the lifespan of either wild-type or *fog-2* females (*Figure 1F–I*). Together, these data suggest that self-sperm protects against brief mating-induced death in a manner that is independent of both male sperm and pheromone effects, and acts through the seminal fluid/IIS pathway.

### Brief mating-induced death acts through the IIS/FOXO seminal fluid pathway

We previously showed that Insulin-IGF-1 signaling (IIS) mediates seminal fluid-induced killing by inducing DAF-16/FOXO nuclear exit (*Shi and Murphy, 2014*). We find that *daf-16* mutants are also susceptible to brief mating, even when young (*Figure 2A*), suggesting that DAF-16 activity is critical for resistance to brief mating. Long-lived *daf-2*/Insulin receptor mutants, although susceptible to long-term mating, are better protected than wild-type worms on Day 8 of adulthood (*Figure 2B*), exhibiting the opposite phenotype as *daf-16* mutants. Furthermore, DAF-16::GFP in *fog-2* (self-spermless) mutants moves out of the nucleus upon brief mating (*Figure 2C–D*).

### Self-sperm modulate expression of *ins-37*

We reasoned that the factor that provides self-sperm protection against seminal fluid-induced killing should already be present in hermaphrodites prior to mating; therefore, we compared the expression profiles of *fog-2* (self-spermless) and wild-type hermaphrodites at L4, a stage after all self-sperm have been made. Not surprisingly, wild-type worms express more major sperm proteins and other sperm-associated genes than do self-spermless *fog-2* mutants (*Figure 3—figure supplement 1*;

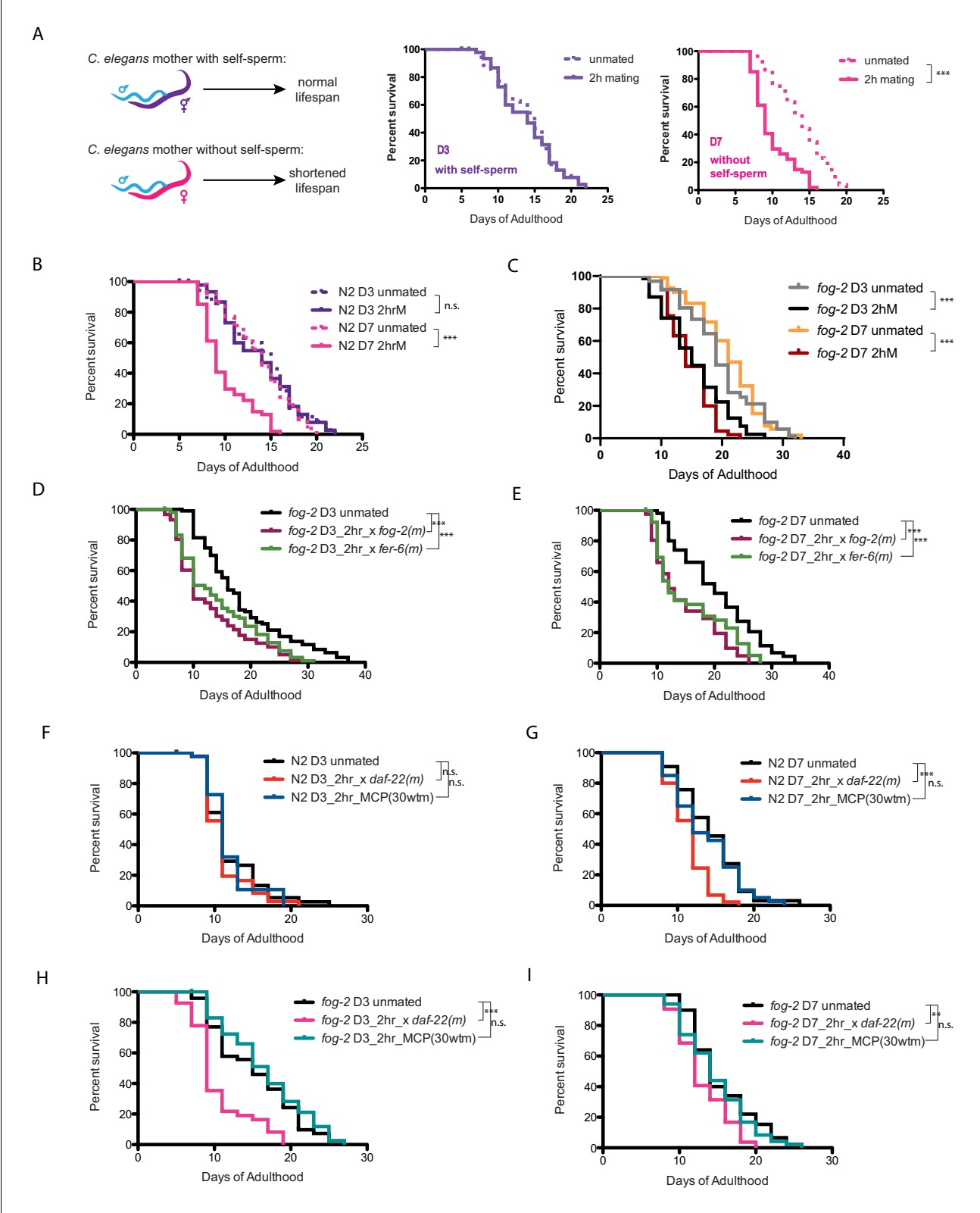

**Figure 1.** Brief mating-induced death is dependent on self-sperm and acts through male seminal fluid. (**A**) Schematic and representative survival curves illustrating survival after a brief (2 hr) mating; *C. elegans* hermaphrodite lifespan is dependent on the presence of self-sperm. (**B**) Wild-type N2 hermaphrodites have reduced lifespan after 2 hr mating on day 7 of adulthood, but have normal lifespan when such brief mating occurs on day 3. Day 3 N2 unmated: 14.1 ± 0.7 days; mated: 13.9 ± 0.6 days, p=0.8481; Day 7 N2 unmated: 13.8 ± 0.5 days; mated: 10.0 ± 0.4 days, p<0.0001. In this study, we

*Figure 1 continued on next page*

*Figure 1 continued*

define day 0 as the beginning of the adulthood for all lifespans. Kaplan-Meier analysis with log-rank (Mantel-Cox) method was performed to compare the lifespans of different groups in this study. See **Supplementary file 1** for all lifespan data summary. (C) Self-spermless *fog-2(q71)* are susceptible to 2 hr mating on both day 3 and day 7 of adulthood. Day three *fog-2* unmated: 19.7 ± 0.7 days; mated: 15.4 ± 0.7 days, p<0.0001; Day seven *fog-2* unmated: 21.0 ± 0.6 days; mated: 14.8 ± 0.5 days, p<0.0001. (D–E) Mating with *fog-2* males (which are functionally wild type) or with *fer-6* males (which lack sperm) leads to a similar magnitude of lifespan decrease in *fog-2* self-spermless hermaphrodites. Day 3 *fog-2* unmated: 18.2 ± 0.8 days; mated with *fog-2* males: 12.6 ± 0.7 days, p<0.0001; mated with *fer-6* males: 14.1 ± 0.7 days, p=0.0002 (compared to unmated). Day 7 *fog-2* unmated: 20.3 ± 1.0 days; mated with *fog-2* males: 14.6 ± 0.9 days, p<0.0001; mated with *fer-6* males: 15.6 ± 1.0 days, p=0.0016 (compared to unmated). (F–G) N2 hermaphrodites are not short-lived after 2 hr exposure to male pheromone-conditioned plates (conditioned by 30 wild-type males for 2 days) either on day 3 or day 7 of adulthood. Only old day 7 N2 hermaphrodites have a shorter lifespan after 2 hr mating with *daf-22* male pheromone production defective males (F). Day 3 N2 unmated: 11.9 ± 0.6 days; mated with *daf-22* males: 11.1 ± 0.4 days, p=0.3058; unmated on male-conditioned plates (MCP; pre-conditioned with 30 males, see Materials and methods for details): 11.7 ± 0.5 days, p=0.9696 (compared to unmated). Day 7 N2 unmated: 14.3 ± 0.7 days; mated with *daf-22* males: 11.4 ± 0.4 days, p=0.0001; unmated on male-conditioned plates (MCP): 13.7±0.7 days, p=0.5984 (compared to unmated). (H–I) Like N2 hermaphrodites, the lifespans of *fog-2* hermaphrodites are not affected by 2 hr male pheromone exposure. However, both young (G) and old (H) *fog-2* hermaphrodites live significantly shorter after brief mating with *daf-22* males. Day 3 *fog-2* unmated: 15.3 ± 0.8 days; mated with *daf-22* males: 10.4 ± 0.6 days, p<0.0001; unmated on male-conditioned plates (MCP): 16.5 ± 0.8 days, p=0.3459 (compared to unmated). Day 7 *fog-2* unmated: 15.5±0.6 days; mated with *daf-22* males: 13.1 ± 0.4 days, p=0.0017; unmated on male-conditioned plates (MCP): 14.7±0.6 days, p=0.4493 (compared to unmated).

DOI: https://doi.org/10.7554/eLife.46413.002

The following figure supplement is available for figure 1:

**Figure supplement 1.** The presence of self-sperm protects the hermaphrodites from brief mating-induced death.

DOI: https://doi.org/10.7554/eLife.46413.003

**Supplementary file 2**). However, of the remaining differences, the conserved insulin-like peptide *ins-37* is strikingly upregulated in wild-type compared to *fog-2* L4 hermaphrodites (**Figure 3A,B**), suggesting that this germline-expressed (**Ebbing et al., 2018**) signaling molecule is only highly expressed when self-sperm are present. The identification of an insulin-like peptide is also consistent with our finding that brief mating utilizes the seminal fluid/IIS pathway (**Figure 1D–E**).

Elimination of a factor that confers protection from seminal-fluid-induced killing should render young, sperm-containing worms susceptible. Indeed, RNAi of *ins-37* in wild-type (N2) worms shortens their lifespan after brief mating (**Figure 3C**, **Figure 3—figure supplement 2B**), despite the presence of self-sperm, suggesting that ins-37 is key for self-sperm-mediated protection from mating.

INS-37 may act as a self-sperm-induced antagonist of the insulin receptor, DAF-2, consistent with a recent report that INS-37 acts a DAF-2 antagonist during development (**Zheng et al., 2018**). Together, our data suggest that self-sperm protect hermaphrodites from brief mating-induced death through their expression of the INS-37 antagonist, which prevents DAF-2/Insulin receptor activation and the subsequent loss of expression of protective genes by the DAF-16/FOXO transcription factor (**Murphy et al., 2003**; **Tepper et al., 2013**).

## TFEB/HLH-30 and TOR signaling mediate self-sperm protection from seminal fluid-induced killing

Our transcriptional data revealed that spermless females differ not only in expression of sperm proteins and *ins-37* but also in genes whose promoters contain the predicted binding motif for the HLH-30/TFEB transcription factor (**Reimand et al., 2007**; **Grove et al., 2009**; **Figure 3—figure supplement 1**), which has been previously implicated in longevity regulation through its control of lipid metabolism and lysosomal autophagy. Mutants that produce an excess of self-sperm (*fem-3(q20)*) (**Rosenquist and Kimble, 1988**) are protected from brief mating-induced death for a longer period of their lifespan than wild-type hermaphrodites (**Booth et al., 2019**), and we found that the HLH-30 binding motif is also enriched in promoters of genes that are up-regulated in (*fem-3(q20)*) hermaphrodites compared to animals with normal levels of sperm (**Figure 3—figure supplement 1**; **Supplementary file 3**).

TFEB/HLH-30 acts downstream of several longevity pathways, notably the TOR pathway (**Lapierre et al., 2013**; **Nakamura et al., 2016**), and DAF-16 and HLH-30 were also recently shown to cooperate under many conditions (**Lin et al., 2018**). To determine whether HLH-30 is required for self-sperm protection, we briefly mated *hlh-30* mutants on Day 3 of adulthood; similar to loss of *ins-*

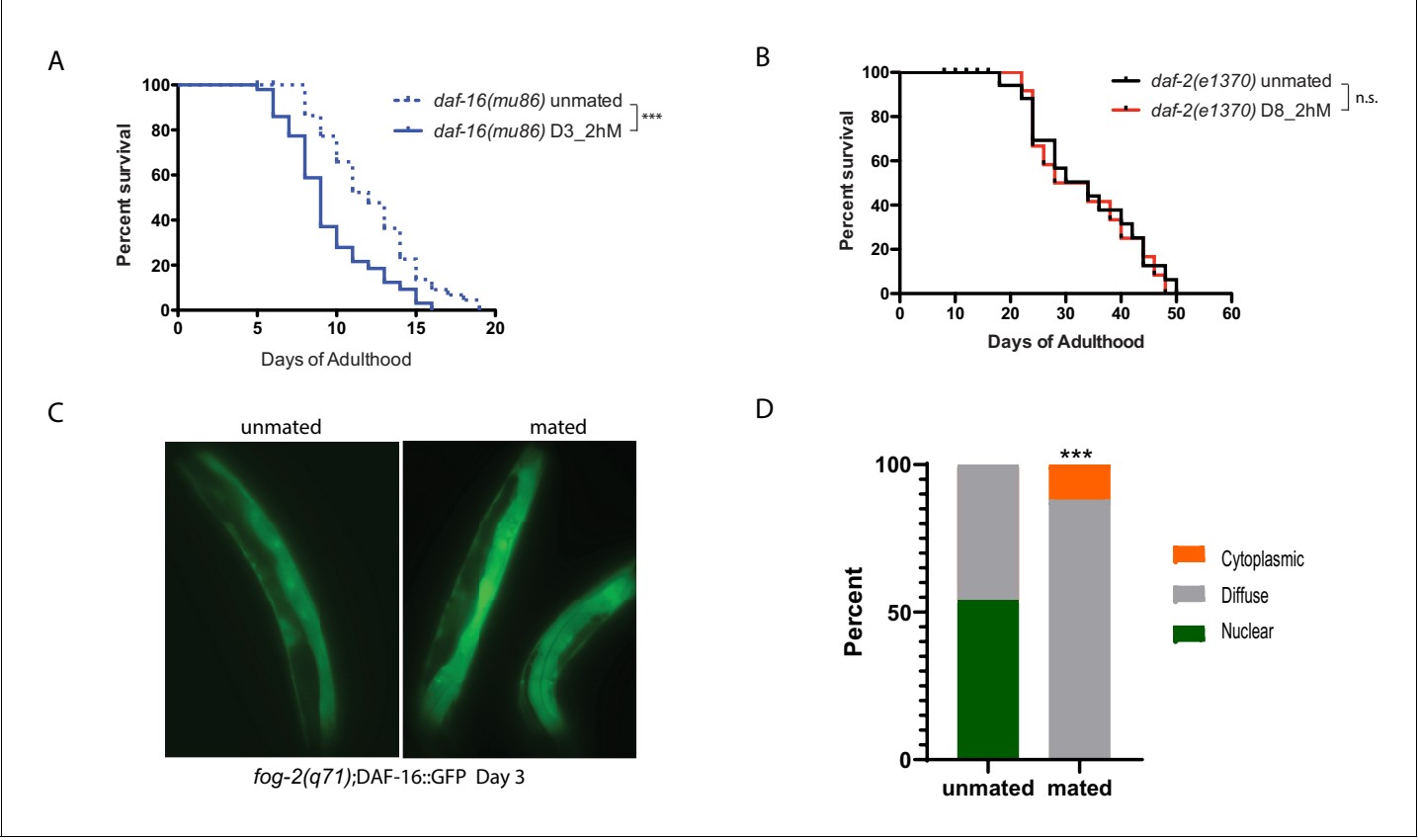

**Figure 2.** IIS/FOXO in brief mating-induced death. (**A**) *daf-16(mu86)* mutants lived shorter after 2hr mating on Day 3 of adulthood. Unmated: 12.2±0.5 days; mated: 9.5±0.5 days, p=0.0003. (**B**) daf-2(e1370) mutants are resistant to 2 hr mating in the absence of self-sperm on Day 8 of adulthood. Unmated: 33.2±2.6 days; mated: 32.7±2.9 days, p=0.7519. (**C–D**) Nuclear DAF-16::GFP in self-spermless *fog-2* hermaphrodites becomes diffuse after mating. (**C**) Representative images of DAF-16::GFP in unmated and mated *fog-2* hermaphrodites (Day 3). (**D**) Quantitation of DAF-16::GFP, p=0.0006. Each worm was assigned a category based on DAF-16::GFP localization. Chi-square test was used to determine the significance.
DOI: https://doi.org/10.7554/eLife.46413.004

37 and *daf-16*, *hlh-30* mutants are susceptible to brief mating even when young, suggesting that self-sperm protection requires HLH-30 activity (*Figure 4A*, *Figure 4—figure supplement 1L*).

To further test the model that TFEB/HLH-30 confers protection, we monitored the localization of HLH-30::GFP (*Craig et al., 2013*) in young, old, and young self-spermless (*fog-2*) animals. We found that HLH-30::GFP is nuclearly localized in the intestine of young animals that have self-sperm, becoming diffuse with age as self-sperm is depleted (*Figure 4B–D*). Fewer self-spermless animals display nuclearly localized HLH-30::GFP, while more animals with excess self-sperm exhibit nuclearly localized HLH-30::GFP (*Figure 4E–F*; note scale). Moreover, loss of *ins-37* reduces nuclear HLH-30::GFP (*Figure 4G–I*), suggesting that INS-37 acts upstream of HLH-30. Our data suggest that DAF-16 and HLH-30 are similarly affected by SF-induced insulin signaling.

The HLH-30::GFP strain overexpresses functional HLH-30 protein, so we wondered whether these animals are better protected from brief mating-induced death; indeed, no lifespan reduction was observed in the HLH-30::GFP strain after mating for 12 hr, a length of mating that shortens WT lifespan (*Figure 4J–K*), suggesting that the additional HLH-30 provides a protective effect. However, longer mating reduces both nuclear HLH-30::GFP localization and lifespan (*Figure 4L*, *Figure 4—figure supplement 1A–K*), suggesting that prolonged mating can eventually overwhelm the protection conferred by HLH-30.

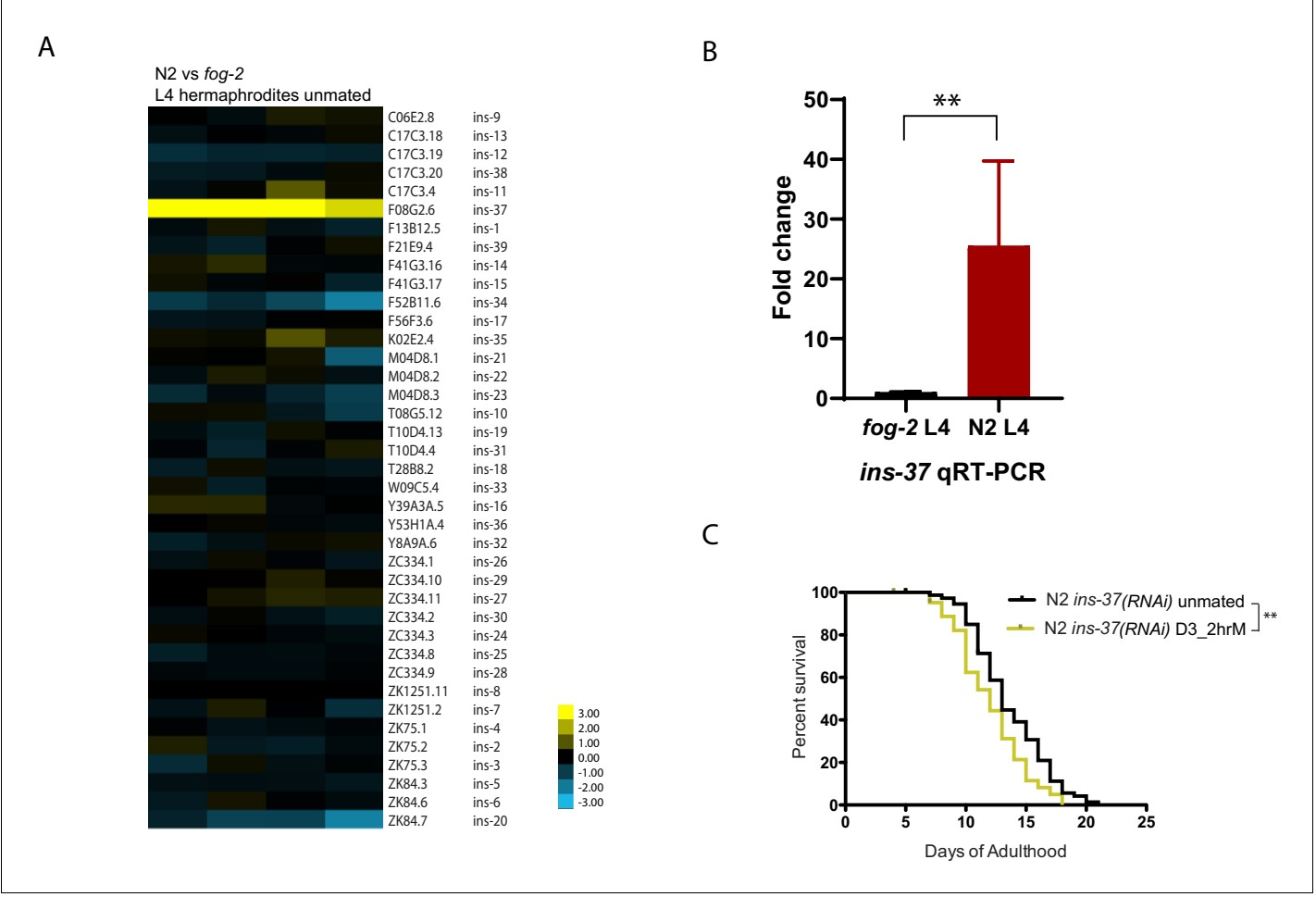

**Figure 3.** INS-37 is required for self-sperm protection from brief mating-induced death. (**A**) Heatmap of all the insulin genes from microarray-based transcriptome comparison between N2 and *fog-2* L4 hermaphrodites. (Four biological replicates) (**B**) qRT-PCR reveals over 25 fold up-regulation of *ins-37* in N2 L4 hermaphrodites compared to *fog-2* L4 hermaphrodites, p=0.0043. Reference gene *pmp-3* was used for normalization. RNA was extracted from three additional biological replicates (different from the sample used for microarray in *Figure 3A*). (**C**) Knocking down *ins-37* in N2 hermaphrodites makes them susceptible to 2 hr mating on day 3 of adulthood. (see *Figure 3—figure supplement 2B* for results of N2 on control RNAi vector) N2 *ins-37* RNAi unmated: 13.6 ± 0.4 days; N2 *ins-37* RNAi mated: 12.0 ± 0.4 days, p=0.0040.

DOI: https://doi.org/10.7554/eLife.46413.005

The following figure supplements are available for figure 3:

**Figure supplement 1.** Transcriptomic comparison between self-spermless, wild type, and excess self-sperm mutants reveals HLH-30 binding motif.
DOI: https://doi.org/10.7554/eLife.46413.006

**Figure supplement 2.** *Ins-7* is highly up-regulated in L4 hermaphrodites with self-sperm.
DOI: https://doi.org/10.7554/eLife.46413.007

## TOR promotes seminal-fluid-induced killing

TOR signaling has been implicated in longevity regulation from yeast through humans, including *C. elegans* (*Kapahi and Hansen, 2010*; *Kaeberlein and Shamieh, 2010*). TOR negatively regulates TFEB/HLH-30's nuclear localization and activity (*Nakamura et al., 2016*). Through genome-wide transcriptional comparisons, we observed that most positive regulators of TOR signaling were modestly downregulated in conditions in which worms are protected from brief mating (*Figure 5A,B*). Therefore, we wondered whether TOR signaling is also involved in longevity regulation in response to mating. Indeed, we found that reduction of the worm TOR homolog, *let-363*, prevented the lifespan decrease caused by brief mating in young spermless (*fog-2*) females (*Figure 5C–D*; *Figure 5—figure supplement 1A–B*). Similarly, RNAi reduction of other TOR pathway components, including *rict-1*, *daf-15*, and *raga-1*, also prevented brief mating-induced death in Day 3 *fog-2* females, and

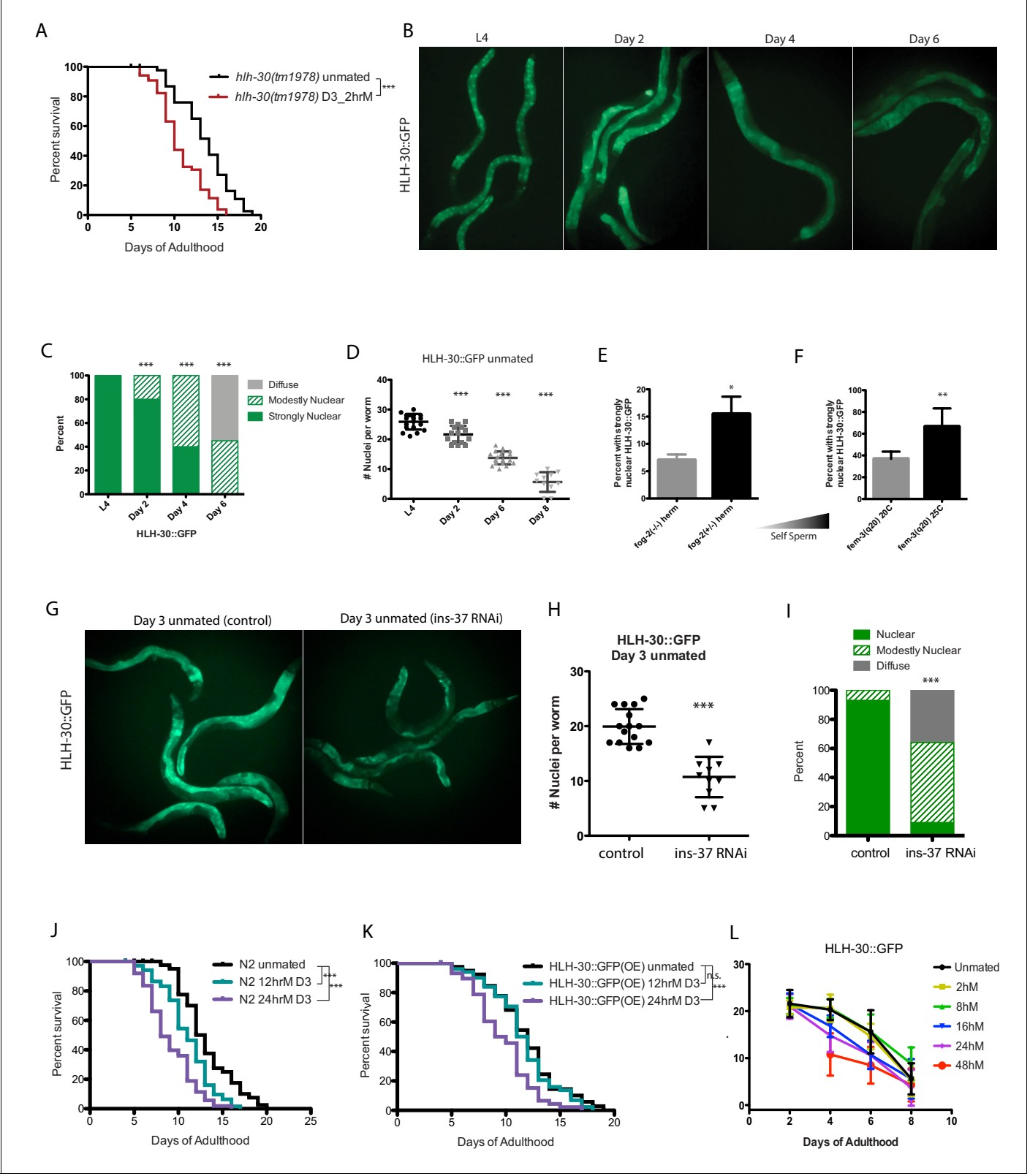

**Figure 4.** HLH-30 protects against brief mating-induced death. (**A**) *hlh-30(tm1978)* mutants live shorter after brief mating on day 3, unlike wild type (N2) hermaphrodites, which are resistant to 2 hr mating on day 3 of adulthood (***Figure 4—figure supplement 1L***). *hlh-30* unmated: 11.7 ± 0.3 days; mated: 9.8 ± 0.3 days, p=0.0002. (**B**) Nuclearly localized HLH-30::GFP becomes diffuse with age. Representative images of HLH-30::GFP (BC11288) in unmated hermaphrodites at various ages. Worms were imaged under free moving conditions; no anesthetic was used, to prevent possible artificial perturbation

*Figure 4 continued on next page*

*Figure 4 continued*

of HLH-30 localization. (**C–D**) Quantitation of HLH-30::GFP in B). In C), each worm was assigned a category based on HLH-30::GFP localization. Chi-square test was used to determine the significance. In D), number of nuclei with nuclearly localized HLH-30::GFP was counted for each worm. The average of each group was compared to L4 using the unpaired t-test. (**E–F**) The amount of self-sperm is positively correlated with the number of worms having strongly nuclearly localized HLH-30::GFP. (Three biological replicates per condition; average was compared by t-test). In F), *fem-3* worms were grown at 25C to induce excess self-sperm phenotype, control *fem-3* were grown at 20 C. (**G–I**) Knockdown of *ins-37* reduces nuclearly localized HLH-30::GFP in unmated hermaphrodites. (**G**) Representative images of day 3 HLH-30::GFP hermaphrodites on control RNAi (left) and *ins-37(RNAi)* (right). (**H–I**) Quantitation of HLH-30::GFP in G), same method used in C-D. (**J–K**) HLH-30::GFP strain is more resistant to longer mating, compared to wild-type (N2) hermaphrodites. Wild-type are already susceptible to 12 hr mating (J), whereas HLH-30::GFP worms are resistant (K). 24 hr mating reduces lifespan of both strains. N2 unmated: 13.2 ± 0.5 days; 12 hr mated: 11.0 ± 0.3 days, p=0.0005; 24 hr mated: 9.0 ± 0.4 days, p<0.0001 (compared to unmated). HLH-30::GFP unmated: 11.9 ± 0.4 days; 12 hr mated: 11.6 ± 0.4 days, p=0.4471; 24 hr mated: 9.7 ± 0.4 days, p=0.0001 (compared to unmated). (**L**) Summary of nuclear HLH-30::GFP quantitation of worms with age and after mating for various time on day 3. (See *Figure 4—figure supplement 1A–K* for more detailed information).

DOI: https://doi.org/10.7554/eLife.46413.008

The following figure supplements are available for figure 4:

**Figure supplement 1.** Quantitation of HLH-30::GFP in worms at different ages and after mating for various lengths of time.

DOI: https://doi.org/10.7554/eLife.46413.009

**Figure supplement 2.** Mating also decreases nuclear HLH-30::GFP in *fog-2* self-spermless hermaphrodites.

DOI: https://doi.org/10.7554/eLife.46413.010

Day 7 (sperm-exhausted) *raga-1* mutant hermaphrodites were also immune to brief mating. Together, these results suggest that downregulation of the TOR pathway is protective (*Figure 5E–F*; *Figure 5—figure supplement 1C–F*).

To test whether TOR's effects are mediated by HLH-30, we subjected *hlh-30(tm1978);let-363 (RNAi)* animals to brief mating, and found that these animals died faster (*Figure 5G*, *Figure 5—figure supplement 1G–H*), suggesting that HLH-30 acts downstream of TOR signaling to regulate brief mating-induced death. Furthermore, reduction of *TOR/let-363* prevents cytoplasmic localization of HLH-30 upon mating (*Figure 5H*; *Figure 5—figure supplement 2A–C*), consistent with its prevention of brief mating-induced killing (*Figure 5C*). Together, these results suggest that TOR promotes SF-mediated killing through its inhibition of TFEB/HLH-30 nuclear localization and activation.

## PQM-1 is required for seminal-fluid-induced killing

Previously, we found that the Zn-finger transcription factor PQM-1 acts downstream of IIS and antagonistically to DAF-16/FOXO nuclear localization under normal conditions (*Tepper et al., 2013*). *pqm-1* mutant males are immune to both mating-induced and male pheromone-induced death (*Shi et al., 2017*), but PQM-1's role in hermaphroditic death was unknown. Unlike *daf-16* mutants, which are susceptible to brief mating even as young adults (*Figure 2A*), *pqm-1* mutants are impervious to brief mating (*Figure 6A*); in fact, they are even resistant to long-term (24 hr) mating (*Figure 6B*, *Figure 6—figure supplement 1A–B*). While DAF-16 leaves the nucleus upon mating (*Shi and Murphy, 2014*) (*Figure 2C–D*), PQM-1 translocates into the nucleus (*Figure 6C–E*). These results suggest that PQM-1 activity is key for seminal fluid-induced killing, acting in an opposite manner to DAF-16/FOXO.

## PQM-1 antagonizes HLH-30 upon mating

We hypothesized that PQM-1 might interact with HLH-30 to regulate SF-mediated killing. *pqm-1 (RNAi)* appears to have no effect on the localization of HLH-30::GFP in unmated day 3 adult worms (*Figure 5—figure supplement 2A–B*). By contrast, under mated conditions (where HLH-30 normally traffics out of the nucleus, 'control mated' in *Figure 6F*), reduction of *pqm-1* prevented this translocation (*Figure 6F*). Consistent with a role in the regulation of HLH-30 localization after mating, reduction of *pqm-1* also prevented the mating-induced lifespan decrease exhibited by *hlh-30* reduction (*Figure 6G*, *Figure 6—figure supplement 1C*). Together, these results suggest that PQM-1 normally acts downstream of the seminal fluid/IIS pathway and antagonistically to HLH-30, and that PQM-1 activity is deleterious to the mother when induced upon the transfer of seminal fluid upon mating.

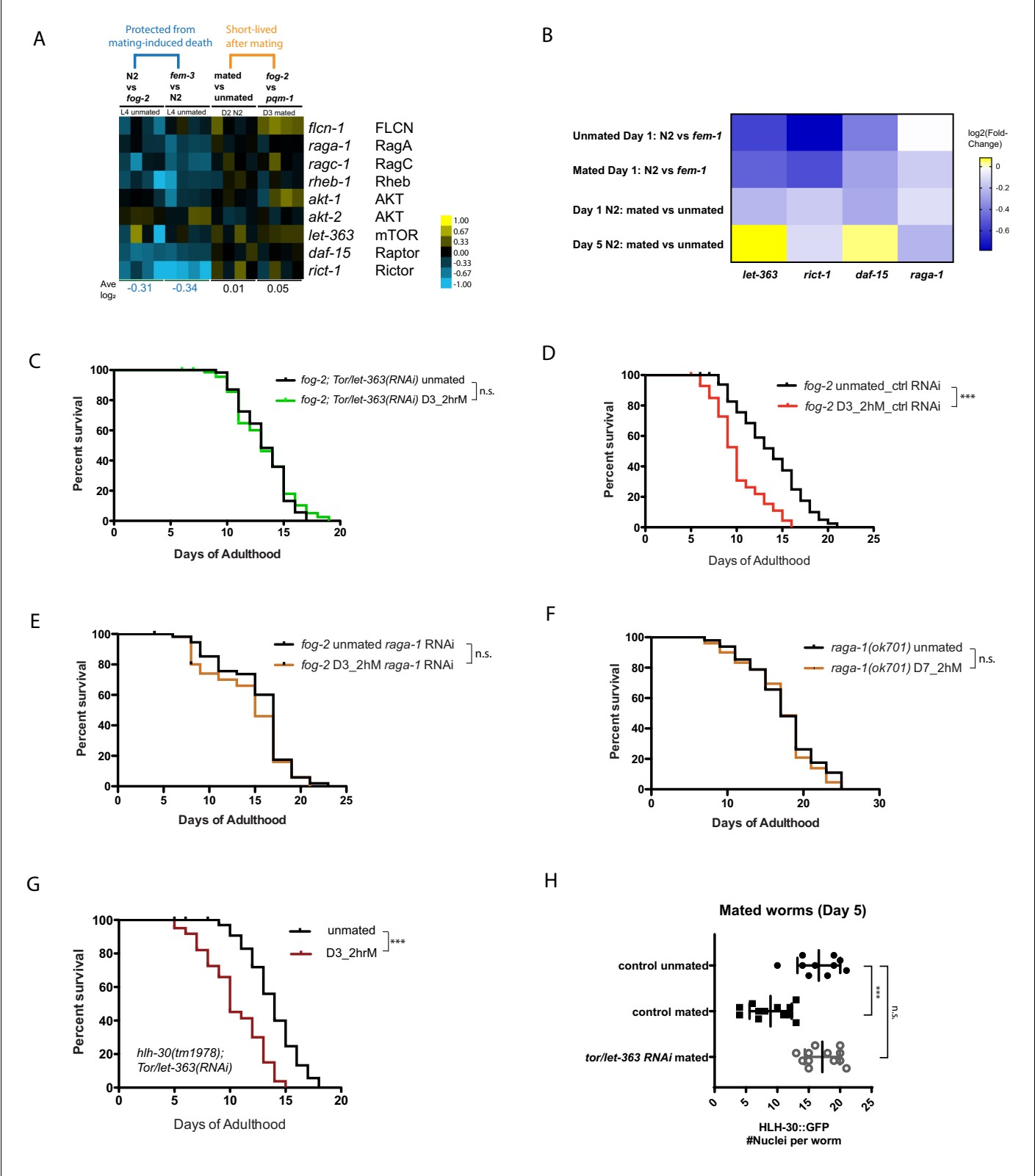

**Figure 5.** TOR signaling components promote Seminal Fluid-induced killing/brief mating-induced death. (**A**) Most positive regulators of TOR signaling (e.g., Rictor/*rict-1*) were modestly downregulated in conditions in which worms have self-sperm and are protected from brief mating-induced death (left block, blue), but rise in mated or short-lived conditions (right block). (**B**) Independent RNA-seq confirmed the modest downregulation of TOR signaling components in worms that are protected from brief mating-induced death. Note the first two rows are conditions in which the hermaphrodites have

*Figure 5 continued on next page*

*Figure 5 continued*

self-sperm protection against brief mating. (C) Knockdown of *tor/let-363* in self-spermless *fog-2* protects the hermaphrodites from SF killing. *fog-2;TOR/let-363(RNAi)* unmated: 13.3 ± 0.3 days; mated: 13.2 ± 0.4 days, p=0.7368. (D) Control of *Figure 5C*: *fog-2* on vector control RNAi unmated: 13.7 ± 0.6 days; mated: 10.1 ± 0.4 days, p<0.0001. (E) *fog-2;raga-1(RNAi)* unmated: 15.3 ± 0.5 days; mated: 14.3 ± 0.6 days, p=0.2378. (F) Day 7 self-sperm depleted *raga-1(ok701)* are also resistant to brief 2 hr mating. Unmated: 18.5 ± 0.7 days; mated: 18.1 ± 0.7 days, p=0.5833. (G) *TOR/let-363* knockdown in *hlh-30(tm1978)* mutants fails to protect the hermaphrodites from brief mating-induced death. *hlh-30;TOR/let-363(RNAi)* unmated: 13.8 ± 0.3 days; mated: 10.5 ± 0.4 days, p<0.0001. (H) Quantitation of nuclear HLH-30::GFP in mated hermaphrodites treated with *let-363* RNAi.
DOI: https://doi.org/10.7554/eLife.46413.011

The following figure supplements are available for figure 5:

**Figure supplement 1.** TOR signaling mediates self-sperm protection from brief mating.
DOI: https://doi.org/10.7554/eLife.46413.012

**Figure supplement 2.** Insulin and mTOR signaling pathways influence HLH-30 and DAF-16 localization in mated hermaphrodites.
DOI: https://doi.org/10.7554/eLife.46413.013

## Insulin-like peptide agonists mediate SF killing

Because self-sperm-mediated protection from SF-induced death correlated with the expression of an antagonist of DAF-2, *ins-37*, we reasoned that a competing, agonist insulin signal might induce the IIS pathway upon Seminal Fluid transfer. Indeed, upon mating, *fog-2* hermaphrodites (which are killed by brief mating) express strikingly higher levels of *ins-8* than do *pqm-1* hermaphrodites, which are not affected by brief mating (*Figure 7A*; *Supplementary file 4*). We confirmed the transcriptome analysis using a transcriptional reporter: mating significantly increased the expression level of *ins-8p::gfp*, particularly in the anterior part of the intestine (*Figure 7B*). INS-7, which has been previously shown to act as a DAF-2 agonist that shortens lifespan (*Murphy et al., 2003*; *Murphy et al., 2007*), and functions in a feed-forward loop to reduce DAF-2 activity (*Murphy et al., 2007*), is also consistently induced but to a lesser degree. We hypothesized that INS-8 might act as a DAF-2 agonist under mating conditions, and thus might induce seminal-fluid-mediated death; if true, reduction of *ins-8* should prevent SF-induced death, even in self-spermless animals. Indeed, while brief mating of Day three *fog-2* females with wild-type males shortens lifespan (*Figure 7E*), knockdown of *ins-8* (*Figure 7C*) or *ins-7* (*Figure 7D*) blocked this effect. Similar to loss of *pqm-1* and *TOR/let-363*, reduction of *ins-7* and *ins-8* prevented HLH-30 cytoplasmic localization after mating (*Figure 7F*, *Figure 5—figure supplement 2C*). Thus, INS-8 and INS-7 activity promote seminal fluid-induced killing and, like PQM-1 and TOR, act upstream of HLH-30.

## Discussion

### A model of self-sperm protection from Seminal Fluid-induced killing

The presence of self-sperm in hermaphroditic mothers slows the deleterious effects of mating with males, whose seminal fluid pathway 'hijacks' the mother's longevity and stress protection system by altering the activity of three important transcription factors, DAF-16 (*Shi and Murphy, 2014*), and as we show here, PQM-1 and HLH-30. The presence of self-sperm staves off mating effects by inducing the expression of the insulin receptor antagonist, INS-37, which in turn prevents DAF-2/insulin signaling and retains DAF-16 and HLH-30 in the nucleus, and PQM-1 in the cytoplasm. These key transcription factors are regulated by the DAF-2/insulin signaling pathway via a specific insulin-like peptide agonist (INS-8) and antagonist (INS-37) (*Figure 8*), and in HLH-30's case, also by the nutrient sensor TOR/LET-363. INS-8 and INS-7 expression is upregulated in mated mothers and inducing seminal-fluid-mediated killing through activation of DAF-2 and subsequent inhibition of DAF-16 (*Shi and Murphy, 2014*; *Lin et al., 2001*) and HLH-30 nuclear localization and activation. INS-7 induction in the intestine and its feed-forward effect on DAF-2 coordinates the IIS pathway across the entire animal (*Murphy et al., 2007*), which causes rapid deceleration of stress responsive pathways. PQM-1 appears to drive HLH-30 out of the nucleus in an antagonistic manner, as PQM-1 does with DAF-16/FOXO (*Tepper et al., 2013*). Like HLH-30, PQM-1 is negatively regulated by TOR signaling (*Dowen et al., 2016*), but since PQM-1's role is detrimental, the loss of each of these factors prevents mating-induced death (*Figure 8*). Together, this network of signaling pathways and transcription factors coordinates both the resistance and susceptibility to mating-induced death.

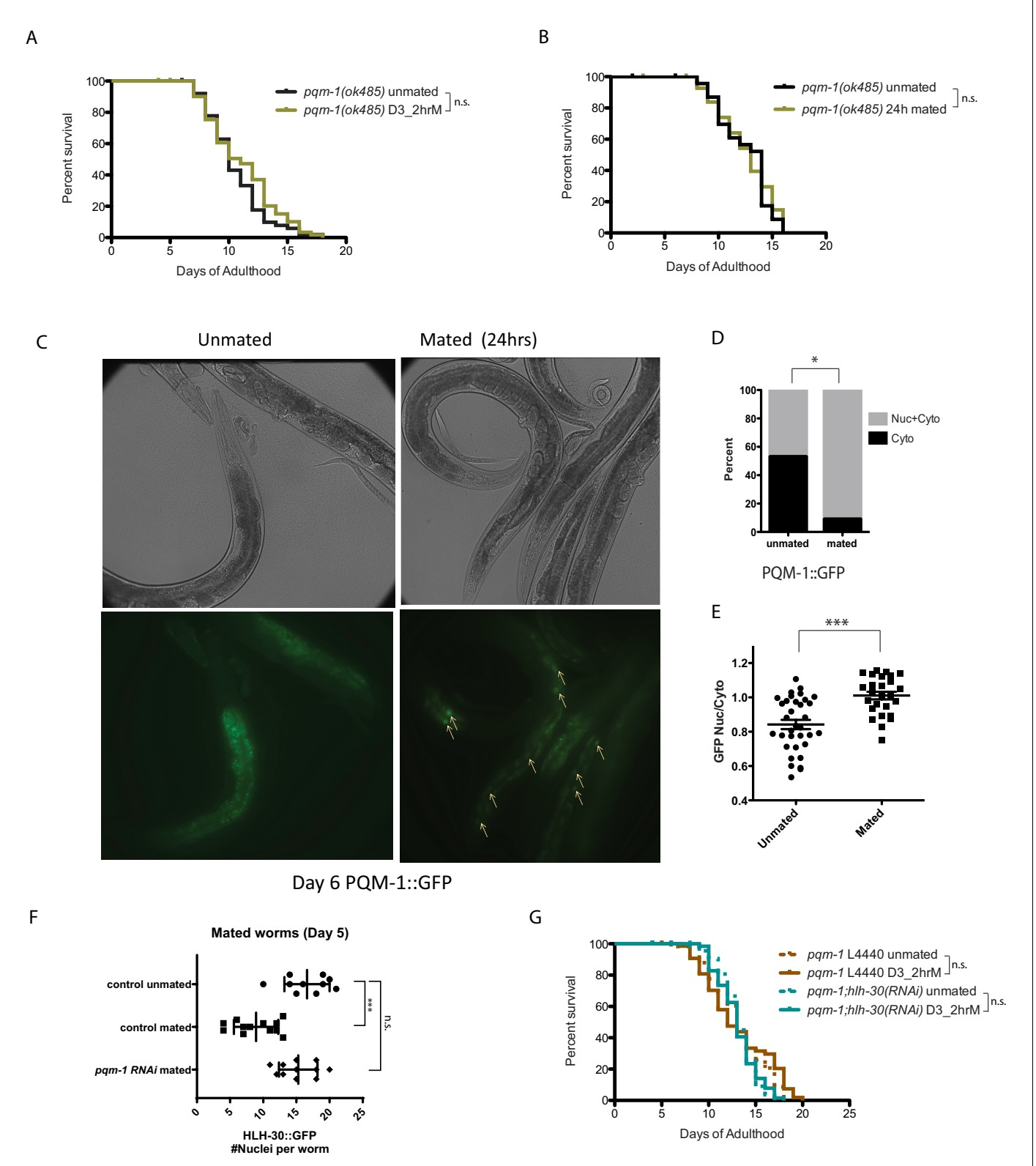

**Figure 6.** PQM-1 is required for SF and brief mating-induced killing. (A–B) *pqm-1(ok485)* mutant hermaphrodites are resistant to short term (2 hr) mating-induced death (A), as well as long term (24 hr) mating (B) 2 hr *pqm-1* unmated: 10.5 ± 0.3 days; mated: 11.1 ± 0.4 days, p=0.2070. 24 hr *pqm-1* unmated: 12.5 ± 0.5 days; mated: 12.5 ± 0.5 days, p=0.8407. (See **Figure 6—figure supplement 1A–B** for more controls) (C) Representative images of PQM-1::GFP hermaphrodites. Left: unmated *pqm-1* control, Right: mated *pqm-1* (mated on day 1 of adulthood for 24 hr). Images are taken on day 6 of

*Figure 6 continued on next page*

*Figure 6 continued*

adulthood. White arrows indicate nuclearly localized PQM-1::GFP. (D–E) Quantitation of PQM-1::GFP localization. (D) Non-parametric comparison; E) nuclear and cytoplasmic GFP intensity was measured in the most anterior nuclei (easily recognizable) of the intestine for worms in each group. Averages were compared using a t-test. (F) Knocking down *pqm-1* restores nuclearly localized HLH-30 in mated hermaphrodites. Quantitation of nuclear HLH-30:: GFP in mated hermaphrodites treated with *pqm-1* RNAi (same as *Figure 5H*). (G) Knocking down *pqm-1* blocks the lifespan decrease normally induced by 2 hr mating in *hlh-30(tm1978)* mutants. *pqm-1* on vector control RNAi: unmated:13.2 ± 0.4 days, mated: 13.1 ± 0.5 days, p=0.4169. *pqm-1;hlh-30 (RNAi):* unmated:13.2 ± 0.3 days, mated: 13.1 ± 0.3 days, p=0.9161.

DOI: https://doi.org/10.7554/eLife.46413.014

The following figure supplement is available for figure 6:

**Figure supplement 1.** PQM-1 is required for SF killing.

DOI: https://doi.org/10.7554/eLife.46413.015

The ability of hermaphroditic mothers to slow mating-induced decline as long as she has her own self-sperm suggests that hermaphrodites have developed strategies to combat the deleterious effects of mating with males, staving off early death until mating becomes beneficial to the mother. While early mating is detrimental to the mothers, late mating (i.e., after all self-sperm are utilized) is beneficial, as it allows the mother to use her remaining oocytes. Moreover, the involvement of TOR suggests that nutritional status may also influence mating-induced death. Thus, a dynamic system that is responsive to changing sperm levels and nutrient signals may allow such a shift. The expression of a DAF-2 antagonist, INS-37, allows hermaphrodites to resist and delay *C. elegans'* main mechanism of competition with other males, seminal fluid-induced killing, until self-sperm are exhausted, increasing the chances that the mother will survive through the end of her reproductive period. Such strategies, coupled with the increased attraction of males that accompany the depletion of self-sperm (*Morsci et al., 2011*), enable hermaphrodites to delay both the odds of mating, through behavioral changes (*Morsci et al., 2011*) and the detrimental effects of mating (through molecular signaling), before the full utilization of their own sperm. This increases the chance that they will propagate their own genome preferentially before succumbing to mating. The balance shifts in the male's favor as the hermaphrodite's sperm are depleted, causing her death shortly after producing his progeny, and preventing further mating by other males. This insulin peptide-based arms race regulates a transition from preventing to promoting the activation of the insulin signaling pathway, resulting first in protection from death during early reproduction, and then accelerated death of the mother immediately after reproduction. Together, these signals tune and optimize reproduction for the hermaphroditic mother, even when confronted with deleterious effects from males.

## Materials and methods

### Strains

| Strains | Source |
| --- | --- |
| N2 | CGC |
| CF1038: *daf−16(mu86) I* | CGC |
| BA6: *fer−6(hc6) I* | CGC |
| CB4108: *fog−2(q71) V* | CGC |
| BA17: *fem−1(hc17) IV* | CGC |
| JK816: *fem−3(q20) IV* | CGC |
| DR476: *daf−22(m130) II* | CGC |
| RB711: *pqm−1(ok485) II* | CGC |
| JIN1375: *hlh−30(tm1978) IV* | CGC |
| BC11288: *dpy−5(e907) I; sEx11288 [rCes W02C12.3b::GFP + pCeh361]* | CGC |

*Continued on next page*

*Continued*

| Strains | Source |
|---|---|
| OP201: *unc−119(tm4063) III; wgIs201 [pqm−1::TY1::EGFP::3xFLAG(92C12)+unc−119(+)]* | CGC |
| HT1704: *unc−119(ed3) III; wwEx67[ins−8 p::GFP + unc−119(+)]* | CGC |
| TJ356: *zIs356[daf-16p::daf-16a/b::GFP + rol-6(su1006)]* | CGC |
| CQ777: *fog-2(q71); zIs356 [daf-16p::daf-16a/b::GFP + rol-6(su1006)]* (TJ356 crossed into CB4108) | This study |

## Experimental methods

### Mating setup

60 mm NGM plates were used to set up group mating. Each 60 mm NGM plate was seeded with OP50 to make a ~ 3 cm diameter bacterial lawn 2 days before mating. About 50 hermaphrodites and 150 young (day 1 − day 2 of adulthood) males were transferred onto the plate. Two hours later (for all the brief mating lifespan assays) or 12/24 hr later (for long term mating assays), the hermaphrodites were transferred onto newly seeded 60 mm NGM plates in the absence of males for lifespan assays.

### Lifespan assay

Lifespan assays were performed at 20C. No FUdR was added to the plates. About 25 synchronized Day 1 hermaphrodites were transferred onto each plate. The hermaphrodites were transferred daily onto new seeded plates in the first week of the lifespan assay. Afterwards, they were transferred once every 2 days. Group mating was set up at the beginning of Day 3 of adulthood (or as noted in text) for brief mating lifespan assays. When RNAi was used in lifespan assay, RNAi treatment always started from eggs for all the experiments in this study. Worms were moved onto standard mating plates (NGM seeded with OP50, no RNAi is included) for 2 hr mating to eliminate any potential influence on male mating efficiency caused by RNAi. Then, worms were moved back onto RNAi plates for the remaining of lifespan assays. Kaplan−Meier analysis with log−rank (Mantel−Cox) method was performed to compare the lifespans of different groups. 'Bagged' worms were censored on day of the event.

### Male-conditioned plates (MCP) setup

Male−conditioned plates for lifespan assays were prepared similar to the previous description (*Maures et al., 2014*). Briefly, 60 µl of OP50 was dropped onto each 35 mm NGM plate to make a bacterial lawn of ~25 mm diameter. 30 young Day 1 wild−type males (fog−two males) were transferred onto each plate. Two days later, they were removed and 25 hermaphrodites for lifespan assays were immediately transferred onto these male-conditioned plates (MCP). These MCPs were prepared throughout the course of the lifespan assays to ensure fresh MCP plates ere available.

## GFP fluorescent worm imaging and quantitation

### HLH−30::GFP (BC11288)

Multiple HLH−30::GFP strains are available at CGC. Since we need to mate the worms and monitor their post−mating lifespans, strains with roller markers are not ideal for our purpose, because roller phenotype could potentially affect the mating efficiency especially in the brief mating set up. Therefore, we chose BC11288, in which gfp was inserted after the start codon of the longest *hlh−30* transcript. Another advantage of BC11288 is that its fluorescence is very strong, allowing us to directly image the freely moving worms under the scope, which could reveal HLH−30 localization more accurately, since mounting−induced stress could make HLH−30 more nuclear (*Lapierre et al., 2013*). 1–2 min of videos were taken using iphone connected directly with Leika M205 FA. Worms in focus and with proper exposure were scored in two ways: i) each worm was assigned a category based on HLH−30::GFP localization (strongly nuclear, weakly nuclear, and diffuse); a Chi−square test was used to determine the significance between different groups; ii) how many recognizable intestinal

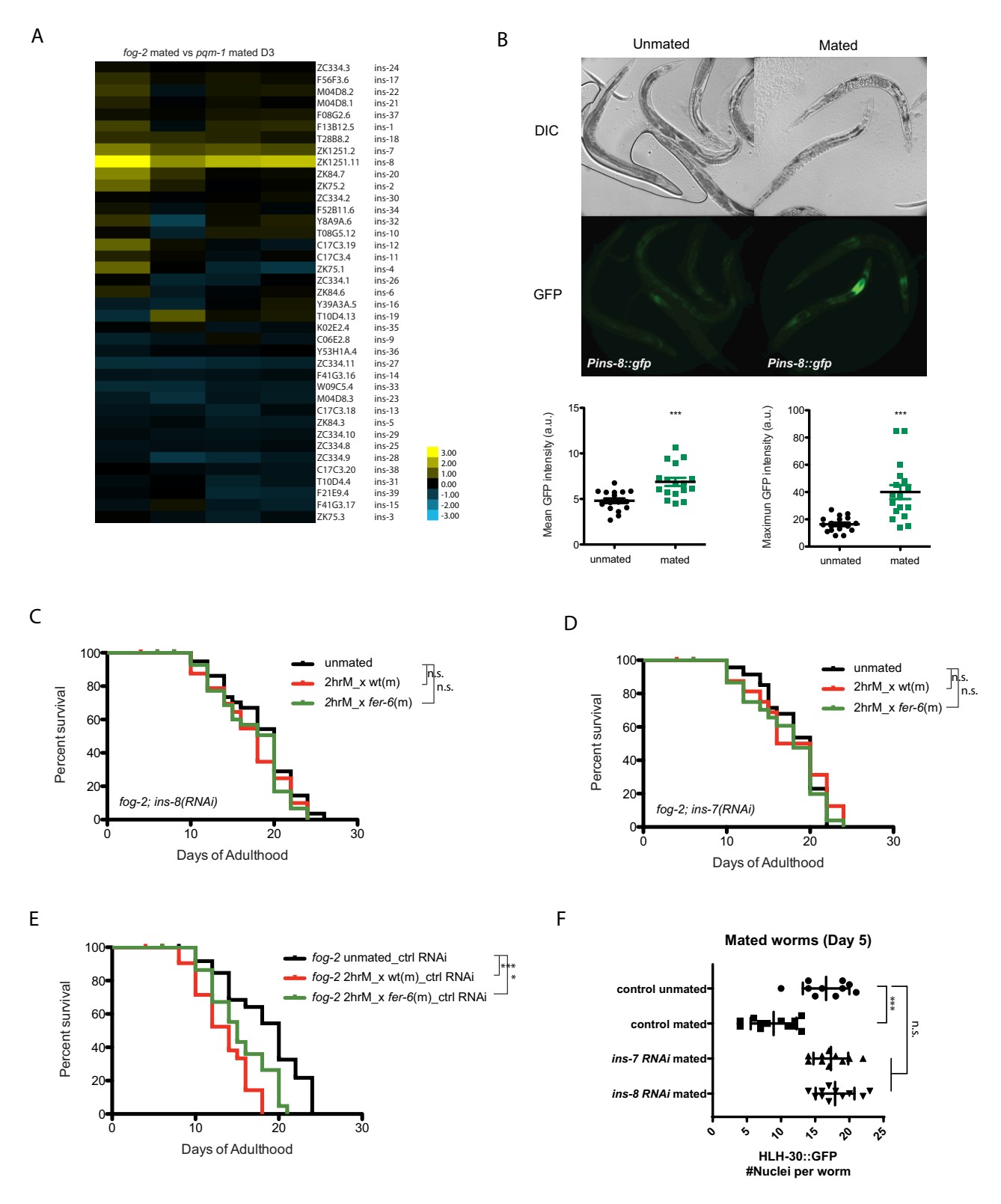

**Figure 7.** Insulins mediate seminal fluid killing. (**A**) Heatmap of insulin gene expression comparisons between *fog-2* mated worms (short-lived after mating) and *pqm-1* mated worms (no lifespan decrease after mating). (**B**) Both mean and maximum *Pins-8::gfp* expression increased in mated worms (right), correlating with gene expression data (**A**). (**C, D**) Knockdown of *ins-8* (**C**) or *ins-7* (**D**) protects hermaphrodites from seminal fluid killing. (**C**) *fog-2; ins-8(RNAi)* unmated: 18.4 ± 0.7 days; mated with *fog-2* males: 17.2 ± 0.9 days, p=0.2879; mated with *fer-6* males: 17.3 ± 0.7 days, p=0.2582 (compared

*Figure 7 continued on next page*

*Figure 7 continued*

to unmated). (D) *fog-2;ins-7(RNAi)* unmated: 18.2 ± 0.6 days; mated with *fog-2* males: 17.7 ± 1.1 days, p=0.9380; mated with *fer-6* males: 17.2 ± 0.7 days, p=0.3994 (compared to unmated). (E) Controls for *Figure 7C–D*. *fog-2* on vector control RNAi unmated: 18.4 ± 0.9 days; mated with *fog-2* males: 13.3 ± 0.7 days, p<0.0001; mated with *fer-6* males: 15.4 ± 0.6 days, p=0.0032 (compared to unmated). (F) Knocking down *ins-7* and *ins-8* also restores nuclearly localized HLH-30 in mated hermaphrodites.

DOI: https://doi.org/10.7554/eLife.46413.016

nucleus can be counted. The average number of intestinal nuclei with nuclearly localized HLH−30:: GFP was compared with each other using the unpaired t−test. When HLH-30::GFP was assayed in the mutant background, the following cross strategy was used: HLH-30::GFP hermaphrodites were crossed with mutant males (i.e. *fog-2*), F1 males with positive GFP signal were crossed with mutant *fog-2* hermaphrodites again, hermaphrodites from F2 generation were directly used in the assay. Half of the were *fog-2* homozygous mutant, half of them only have one copy of - mutant allele, thus have functional self-sperm.

## PQM−1::GFP (OP201)

Worms were mated on day 3 of adulthood for 24 hr. 20–30 worms of each group were imaged by Nikon Eclipse 90i on day 6 of adulthood. PQM−1::GFP localization was quantified in two ways. i) each worm was assigned a category based on PQM−1::GFP localization (nuclear, cytoplasmic, and diffuse(both nuclear and cytoplasmic)). Chi−square test was used to determine the significance between different groups. ii) the mean gfp intensity of the most anterior intestinal nuclei and the

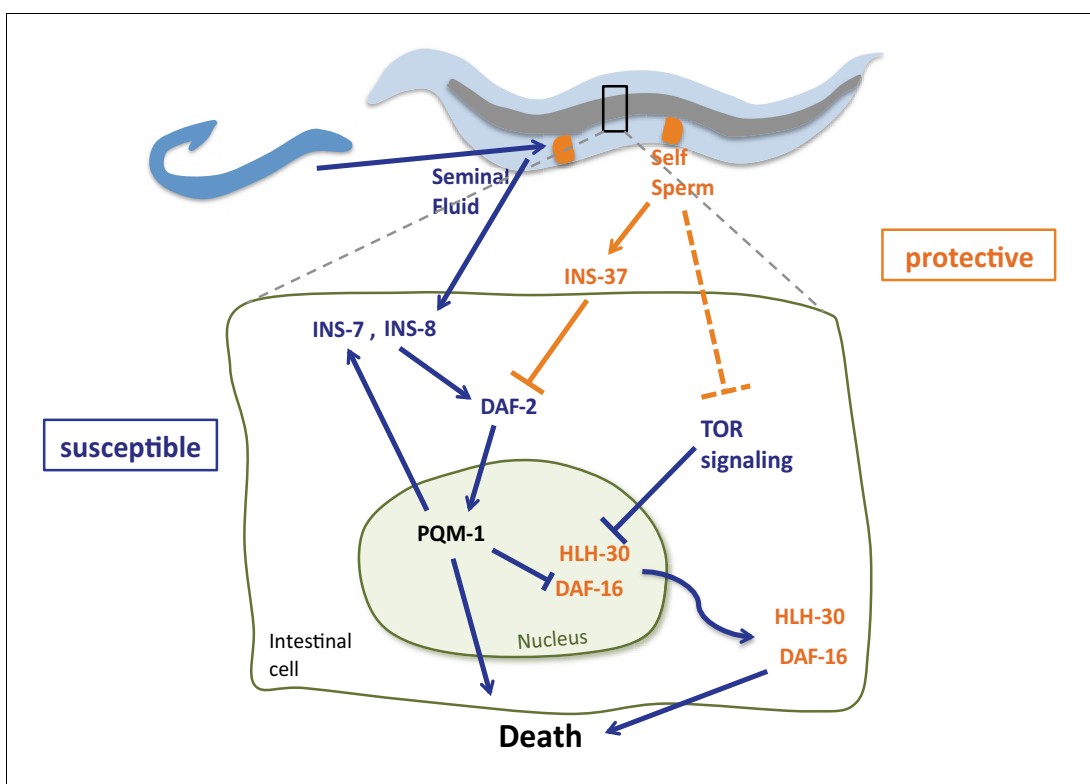

**Figure 8.** Model of self-sperm mediated protection from seminal fluid killing Prior to mating, the presence of self-sperm maintains high *ins-37* expression. INS-37 antagonizes DAF-2 activity and promotes DAF-16 and HLH-30 maintenance in the nucleus. Self-sperm also inhibits TOR signaling, which regulates HLH-30 nuclear localization. Seminal fluid transfer upon mating increases expression of *ins-8* and *ins-7*; these agonists activate DAF-2, promoting PQM-1 nuclear localization and DAF-16 and HLH-30 nuclear exit, resulting in premature death. INS-7 acts in a feed-forward loop (*Murphy et al., 2007*), further accelerating DAF-2 activation.

DOI: https://doi.org/10.7554/eLife.46413.017

nearby cytoplasmic region was measured by image J (DIC picture of the same worm was used to determine the position and the size of the nuclei, then switch to GFP picture to measure gfp intensity. The same outline of the nucleus was used and moved to the cytoplasmic region next to the nucleus, and the gfp intensity of this cytoplasmic region was measured.) The Nuc/Cyto ratio was calculated using the two values. T−test analysis was performed to compare the Nuc/Cyto PQM−1:: GFP intensity of mated and unmated worms.

## DAF-16::GFP (TJ356)
Worms were mated on day 1 of adulthood for 24 hr. 20–30 worms of each group were imaged by Nikon Eclipse 90i next day. DAF−16::GFP localization (nuclear, cytoplasmic, and diffuse(both nuclear and cytoplasmic)). Chi−square test was used to determine the significance between different groups. RNAi treatment started from eggs.

## Pins−8::GFP (HT1704)
Worms were mated on day 3 of adulthood for 24 hr. 20–30 worms of each group were imaged by Nikon Eclipse 90i on day 5 of adulthood. Image J was used to measure the mean and the maximum GFP intensity of the whole body area. T−test analysis was performed to compare the GFP intensity of different group of worms.

## Microarrays
About 200 L4 hermaphrodites were collected for each condition and replicate (*Figure 3A* and *Figure 3—figure supplement 1*). For microarrays comparing mated hermaphrodites (*Figures 5A* and *7A*), hermaprhodites were mated on day 1 of adulthood for 24 hr. 100–200 hermaphrodites were collected on day 3 of adulthood.

Four biological replicates were performed. RNA was extracted by the heat−vortexing method. Two-color Agilent microarrays were used for expression analysis. Significant differentially expressed gene sets were identified using SAM (*Tusher et al., 2001*). g:Profiler was used for GO term analysis and enriched TF motifs prediction (*Reimand et al., 2016*).

Links to the original microarray data:

L4 N2 vs *fog-2(q71)* hermaphrodites: https://puma.princeton.edu/cgi-bin/exptsets/review.pl?exptset_no=7332

L4 *fem-3(q20)* vs N2 hermaphrodites: https://puma.princeton.edu/cgi-bin/exptsets/review.pl?exptset_no=7333

Day three mated *fog-2(q71)* vs mated *pqm-1(ok485)* hermaphrodites: https://puma.princeton.edu/cgi-bin/exptsets/review.pl?exptset_no=7334

## *ins-37* qRT-PCR measurements
About 200 L4 hermaphrodites were collected for each genotype and condition. RNA was extracted by the heat−vortexing method. SuperScript III First-Strand Synthesis (ThermoFisher Scientific) was used to convert extracted RNA to cDNA. qRT-PCR was performed in ViiA seven system (Applied Biosystems) using SYBR Green (Sigma-Aldrich) method. *Ins-37* expression in *fog-2* unmated L4 was normalized to 1. pmp-3 was used as a reference gene.

*Ins-37* primers:
'*ins-37*-qRT-f' 5'-CATCCCGAACCGGATAGACG-3'
'*ins-37*-qRT-r' 5'-GGGACCGGGTGAATTGGATT-3'
Reference gene primers: *pmp-3*
Forward: 5'-AGTTCCGGTTGGATTGGTCC-3'
Reverse: 5'-CCAGCACGATAGAAGGCGAT-3'

## Acknowledgements
L Booth contributed the original observations (*Figure 1A–C*) and RNA-seq data (*Figure 5B*); CS performed all other experiments. We thank L Booth and A Brunet for sharing data pre-publication, the *Caenorhabditis* Genetics Center (CGC) for strains, members of the Murphy laboratory for critically

reading the manuscript, and A Brunet (Stanford) for helpful discussion. This work was supported by an NIH Pioneer award and funding from the Glenn Foundation for Medical Research to CTM.

## Additional information

### Funding

| Funder | Grant reference number | Author |
|---|---|---|
| NIH Office of the Director | Pioneer 1DP1OD020400-01 | Coleen T Murphy |
| Glenn Foundation for Medical Research | | Coleen T Murphy |

The funders had no role in study design, data collection and interpretation, or the decision to submit the work for publication.

### Author contributions

Cheng Shi, Conceptualization, Validation, Investigation, Visualization, Methodology, Writing—original draft, Writing—review and editing; Lauren N Booth, Conceptualization, Validation, Methodology; Coleen T Murphy, Conceptualization, Supervision, Funding acquisition, Writing—original draft, Project administration, Writing—review and editing

### Author ORCIDs

Cheng Shi https://orcid.org/0000-0003-0365-8273
Lauren N Booth https://orcid.org/0000-0003-3072-6235
Coleen T Murphy https://orcid.org/0000-0002-8257-984X

### Decision letter and Author response

Decision letter https://doi.org/10.7554/eLife.46413.030
Author response https://doi.org/10.7554/eLife.46413.031

## Additional files

### Supplementary files

• Supplementary file 1. Lifespan results summary. Summary of all the lifespan results in this study.
DOI: https://doi.org/10.7554/eLife.46413.019

• Supplementary file 2. N2 vs *fog2(q71)* L4 unmated hermaphrodites microarray analysis. List of significantly up- and down-regulated genes in N2 vs *fog2(q71)* L4 unmated hermaphrodites transcriptome comparison identified by Significant Analysis of Microarrays (SAM), and functional enrichment analysis of these genes by g:Profiler.
DOI: https://doi.org/10.7554/eLife.46413.020

• Supplementary file 3. *fem-3(q20)* vs N2 L4 unmated hermaphrodites microarray analysis. List of significantly up- and down-regulated genes in *fem-3(q20)* vs N2 L4 unmated hermaphrodites transcriptome comparison identified by Significant Analysis of Microarrays (SAM), and functional enrichment analysis of these genes by g:Profiler.
DOI: https://doi.org/10.7554/eLife.46413.021

• Supplementary file 4. *fog-2(q71)* vs *pqm-1(ok485)* mated hermaphrodites (Day 3) microarray analysis. List of significantly up- and down-regulated genes in *fog-2(q71)* vs *pqm-1(ok485)* mated hermaphrodites (Day 3) transcriptome comparison identified by Significant Analysis of Microarrays (SAM), and functional enrichment analysis of these genes by g:Profiler.
DOI: https://doi.org/10.7554/eLife.46413.022

• Transparent reporting form
DOI: https://doi.org/10.7554/eLife.46413.023

## Data availability

Microarray data are available at the following links: "L4 fog-2(q71) vs N2 hermaphrodites" https://puma.princeton.edu/cgi-bin/exptsets/review.pl?exptset_no=7332; "L4 fem-3(q20) vs N2 hermaphrodites" https://puma.princeton.edu/cgi-bin/exptsets/review.pl?exptset_no=7333; "D3 mated fog-2(q71) vs pqm-1(ok485) hermaphrodites" https://puma.princeton.edu/cgi-bin/exptsets/review.pl?exptset_no=7334.

The following datasets were generated:

| Author(s) | Year | Dataset title | Dataset URL | Database and Identifier |
|---|---|---|---|---|
| Cheng Shi, Lauren N Booth, Coleen T Murphy | 2019 | L4 fog-2(q71) vs N2 hermaphrodites | https://puma.princeton.edu/cgi-bin/exptsets/review.pl?exptset_no=7332 | Princeton University MicroArray database, 7332 |
| Cheng Shi, Lauren N Booth, Coleen T Murphy | 2019 | L4 fem-3(q20) vs N2 hermaphrodites | https://puma.princeton.edu/cgi-bin/exptsets/review.pl?exptset_no=7333 | Princeton University MicroArray database, 7333 |
| Cheng Shi, Lauren N Booth, Coleen T Murphy | 2019 | D3 mated fog-2(q71) vs pqm-1(ok485) hermaphrodites | https://puma.princeton.edu/cgi-bin/exptsets/review.pl?exptset_no=7334 | Princeton University MicroArray database, 7334 |

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
