## [Decision Letter]

Thank you for submitting your article "Insulin-like peptides and the mTOR-TFEB pathway protect *C. elegans* hermaphrodites from Mating-induced Death" for consideration by *eLife*. Your article has been reviewed by three peer reviewers, one of whom is a member of our Board of Reviewing Editors, and the evaluation has been overseen by Patricia Wittkopp as the Senior Editor. The following individual involved in review of your submission has agreed to reveal his identity: George Sutphin (Reviewer #2).

The reviewers have discussed the reviews with one another and the Reviewing Editor has drafted this decision to help you prepare a revised submission.

All of the reviewers are in agreement that the work described here provides important insights into the mechanisms that control how mating influences health and longevity in worms and that it will be of general interest. However, some concerns were raised regarding experimental design, replication and statistics, and interpretation that the authors are encouraged to address prior to resubmission. In addition, the authors are encouraged to consider a more general readership who may not be experts in the prior literature or *C. elegans* genetic pathways when revising the text.

Essential revisions:

1) Statistics and replication: Few experiments were validated in triplicate in this work and many only included a single biological replicate. At an absolute minimum, all experiments should be performed with duplicate biological replicates and preferably the authors would meet the field standard of triplicate replication.

2) The authors include in their model a strong conclusion that DAF-2 and DAF-16 are mediating the insulin side of the self-sperm protection from mating induced lifespan shortening. No interaction with DAF-2 is explicitly tested in any aspect of this work, and the only experiment examining DAF-16 simply demonstrates that worms lacking *daf-16* are sensitive to brief mating-induced death early in life, while wild-type animals are not. With this single exception, the involvement of DAF-2 and DAF-16 is inferred solely on evidence from interactions with the major genetic players in this story (INS-7, INS-8, INS-37, PMQ-1, HLH-30) in other phenotypic contexts (development, long-term mating, etc). To be clear, the basis for this model is sound, but the interactions have not been confirmed in the context of brief mating-induced death, while involvement of DAF-2 and DAF-16 in this process is strongly stated in the conclusions and final model. One of the compelling reasons for this work and that of Booth et al. is the demonstration that the cost of early-life mating is distinct from late-life or continuous mating (mechanistically because of the presence of self-sperm). Given that some genetic backgrounds appear to have different effects on early-life mating than late-life or continuous mating, inferring relationships between these paradigms is potentially problematic. While experimentally confirming the role of DAF-2 and DAF-16 (e.g. in a manner similar to that presented for HLH-30) would solidify this part of the model, all of these experiments are not absolutely necessary for publication. The reviewers do feel that, at a minimum, the lifespans of *daf-16*(-) mutants should be repeated and a test should be performed for DAF-16::GFP localization. A clearer statement that the role of DAF-2 and DAF-16 are inferred based on previous work on other phenotypes and not directly examined in this work (and similar softening of the final model in Figure 4E) should also be included in the text.

3) Why did the authors focus on *ins-37* in the first place? In subsection “Self-sperm modulate expression of *ins-37*” they discount sperm specific protein, and then decide to look at gene that they state is germline specific and only expressed when self-sperm is present. The most direct conclusion would be that INS-37 is also a sperm specific gene (sperm is part of the germline after all) and to group it in with the others that were already discounted. Figure 2—figure supplement 2A is referenced and seems to have something to do with expression patterns of INS-37, but the authors only provide a reference to another paper and no description of what the acronyms and markers in the figure are supposed to be showing, it is not possible to tell what information is supposed to be conveyed. It is possible that a clear explanation of this figure will clarify the situation, but a better explanation of the selection of INS-37 for follow up is needed.

4) Moreover, *ins-37* gene regulation should be validated either by RTPCR and/or transcriptional reporter (as is the case for ins-8 later in the manuscript). Furthermore, the authors do not show that a brief mating leads to the nuclear exit of DAF-16 as a prolonged mating does. The images showing PQM-1 nuclear translocation are unclear, and rather show an increase in fluorescence expression than an increase in nuclear translocation.

5) The authors show that the transcription factor *hlh-30* is required for protection against mating induced demise. While these data are compelling, the data showing HLH-30 nuclear translocation is not. From the pictures, a decrease of HLH-30 nuclear localization cannot be discerned (Figure 2D, I), and the data on complete nuclear localization of HLH-30 at a young age (Figure 2E) is in contrast to published data (which for the HLH-30::GFP strain from the Irazoqui lab does not indicate nuclear localization is animals at steady state/normal room temperature). Since nuclear localization cannot be evaluated from the images (higher magnification is needed), it is not possible to draw conclusions on the activity of HLH-30 after mating; thus, this data set needs to be revisited. Importantly, what happens to HLH-30 nuclear localization in *fog-2* mutants following mating?

6) The authors find in their RNASeq that components of the mTOR complex are downregulated in animals that are protected against mating-induced demise. These gene expression data should be validated.

7) There is an important control missing from the set of experiments presented in Figure 3C, Figure 3—figure supplement 1B-D. The authors claim that HLH-30 is necessary for LET-363 knockdown to be protective against brief mating on day 3. While they include the *hlh-30* mutant control, the *let-363*(RNAi) control is not shown. In fact, the effect of *let-363* knockdown on mating induced lifespan shortening is only assessed in the *fog-2* background, where the authors also report altered *let-363* expression. It is overall a bit difficult to understand this expected result – if the idea is that mTOR RNAi does not extend lifespan in *fog-2* animals because they have low mTOR pathway activity, how come *fog-2* animals are not long lived (as reported by Kenyon lab)? More rigor and discussion are needed here, and if the authors want to make claims about the functional involvement of the mTOR pathway, the authors should analyze at least one more mTOR pathway mutant, e.g., *raga-1*.

---

## [Author Response]

Essential revisions:1) Statistics and replication: Few experiments were validated in triplicate in this work and many only included a single biological replicate. At an absolute minimum, all experiments should be performed with duplicate biological replicates and preferably the authors would meet the field standard of triplicate replication.

Thank you, we have added more replicates for every experiment (see Supplementary file 1). Our conclusions have not changed with the addition of new experiments.

2) The authors include in their model a strong conclusion that DAF-2 and DAF-16 are mediating the insulin side of the self-sperm protection from mating induced lifespan shortening. No interaction with DAF-2 is explicitly tested in any aspect of this work, and the only experiment examining DAF-16 simply demonstrates that worms lacking daf-16 are sensitive to brief mating-induced death early in life, while wild-type animals are not. With this single exception, the involvement of DAF-2 and DAF-16 is inferred solely on evidence from interactions with the major genetic players in this story (INS-7, INS-8, INS-37, PMQ-1, HLH-30) in other phenotypic contexts (development, long-term mating, etc). To be clear, the basis for this model is sound, but the interactions have not been confirmed in the context of brief mating-induced death, while involvement of DAF-2 and DAF-16 in this process is strongly stated in the conclusions and final model. One of the compelling reasons for this work and that of Booth et al. is the demonstration that the cost of early-life mating is distinct from late-life or continuous mating (mechanistically because of the presence of self-sperm). Given that some genetic backgrounds appear to have different effects on early-life mating than late-life or continuous mating, inferring relationships between these paradigms is potentially problematic. While experimentally confirming the role of DAF-2 and DAF-16 (e.g. in a manner similar to that presented for HLH-30) would solidify this part of the model, all of these experiments are not absolutely necessary for publication. The reviewers do feel that, at a minimum, the lifespans of daf-16(-) mutants should be repeated and a test should be performed for DAF-16::GFP localization. A clearer statement that the role of DAF-2 and DAF-16 are inferred based on previous work on other phenotypes and not directly examined in this work (and similar softening of the final model in Figure 4E) should also be included in the text.

We now have added *daf-2* brief mating data; as expected, *daf-2* mutants are not susceptible to brief mating on Day 8, when self-sperm are depleted (Figure 2B), consistent with our model.

We have also repeated the lifespan of *daf-16(-)* mutants after brief mating, and the result is consistent with our initial result (Figure 2A). Additionally, we have added an analysis of DAF-16::GFP data (Figure 2C-D). DAF-16::GFP is mostly nuclear in unmated *fog-2* hermaphrodites, but becomes diffuse and cytoplasmic after mating, consistent with our model.

3) Why did the authors focus on ins-37 in the first place? In subsection “Self-sperm modulate expression of ins-37” they discount sperm specific protein, and then decide to look at gene that they state is germline specific and only expressed when self-sperm is present. The most direct conclusion would be that INS-37 is also a sperm specific gene (sperm is part of the germline after all) and to group it in with the others that were already discounted. Figure 2—figure supplement 2A is referenced and seems to have something to do with expression patterns of INS-37, but the authors only provide a reference to another paper and no description of what the acronyms and markers in the figure are supposed to be showing, it is not possible to tell what information is supposed to be conveyed. It is possible that a clear explanation of this figure will clarify the situation, but a better explanation of the selection of INS-37 for follow up is needed.

We reasoned that whatever the protective signal is, it should be present when self-sperm are present, prior to any mating event; therefore, we compared unmated wild-type L4s with spermless L4s, anticipating that the relevant molecule should be highly expressed in the former conditions. Furthermore, we had shown that the process is dependent on the transfer of seminal fluid from males into females (now moved to Figure 1), and we further showed that *daf-2/daf-16* insulin signaling pathway regulates brief mating-induced death (*daf-2* mutants are resistant to Day 8 brief mating-induced death, Figure 2A-C). However, the upstream molecule(s) that regulates insulin signaling was unknown in the context of brief mating. Indeed, our expression data show that *ins-37* is very highly differentially expressed in unmated L4s compared to spermless L4s (more than a 20-fold difference, which we have now confirmed by qRT-PCR). Therefore, because we know that the Seminal Fluid pathway regulates the insulin signaling pathway, *ins-37* is an excellent candidate for the protective factor. We have now explained logic this more clearly in the text.

Figure 2—figure supplement 2A has been removed. The purpose was to convey that it is expressed in the germline (and gfp constructs in the germline are often silenced), as was shown by Ebbing, et al., 2018.

4) Moreover, ins-37 gene regulation should be validated either by RTPCR and/or transcriptional reporter (as is the case for ins-8 later in the manuscript). Furthermore, the authors do not show that a brief mating leads to the nuclear exit of DAF-16 as a prolonged mating does. The images showing PQM-1 nuclear translocation are unclear, and rather show an increase in fluorescence expression than an increase in nuclear translocation.

To address the question, we have now performed qRT-PCR of *ins-37* (Figure 2B), which supports our model that *ins-37* expression is greatly increased in animals with self-sperm.

As stated above, we have new data showing that DAF-16::GFP is mostly nuclear in unmated *fog-2* hermaphrodites, but becomes diffuse and cytoplasmic after mating (Figure 2 C,D), consistent with our model.

We have replaced the initial images of PQM-1::GFP with better, clearer images, and we have indicated the nucleus with arrows in the image (Figure 6C).

5) The authors show that the transcription factor hlh-30 is required for protection against mating induced demise. While these data are compelling, the data showing HLH-30 nuclear translocation is not. From the pictures, a decrease of HLH-30 nuclear localization cannot be discerned (Figure 2D, I), and the data on complete nuclear localization of HLH-30 at a young age (Figure 2E) is in contrast to published data (which for the HLH-30::GFP strain from the Irazoqui lab does not indicate nuclear localization is animals at steady state/normal room temperature). Since nuclear localization cannot be evaluated from the images (higher magnification is needed), it is not possible to draw conclusions on the activity of HLH-30 after mating; thus, this data set needs to be revisited. Importantly, what happens to HLH-30 nuclear localization in fog-2 mutants following mating?

We were warned by the lab that sent us the HLH-30 (and we did confirm) that HLH-30::GFP quickly traffics out of the nucleus upon the stress of mounting for microscopy, which limits one’s ability to take as high-magnification images as we normally do with fixed worms. To solve this problem, we took images and videos of freely moving HLH-30::GFP to avoid any inaccuracy caused by microscope mounting-induced localization bias; however, the nuclear localization was clear in our assays. We have now explained these details in the Materials and methods. (Please note that the HLH-30::GFP strain we used for the experiments in the manuscript is BC11288, in which gfp was inserted after the start codon of the longest *hlh-30* transcript (Craig et al., 2013, and on Wormbase, there are quite a few HLH-30::GFP pictures curated by Ian Hope’s lab showing nuclear localization of HLH-30 under steady-state conditions.)

We did notice that the printed versions of our figures after manuscript compression was lower than our original images, so we will work with the journal to be sure that the images in the final version of the paper are as the originals, and that the printed and screen images are equally clear, in case that is the source of the problem.

HLH-30::GFP also became diffuse after mating in *fog-2* mutants (Figure 4—figure supplement 2).

6) The authors find in their RNASeq that components of the mTOR complex are downregulated in animals that are protected against mating-induced demise. These gene expression data should be validated.

We confirmed our microarray results with Lauren Booth’s independent RNA-seq data, which we now show as Figure 5B. Both data sets suggest that worms in the “protective” state have lower expression levels of the mTOR complex components.

7) There is an important control missing from the set of experiments presented in Figure 3C, Figure 3—figure supplement 1B-D. The authors claim that HLH-30 is necessary for LET-363 knockdown to be protective against brief mating on day 3. While they include the hlh-30 mutant control, the let-363(RNAi) control is not shown. In fact, the effect of let-363 knockdown on mating induced lifespan shortening is only assessed in the fog-2 background, where the authors also report altered let-363 expression. It is overall a bit difficult to understand this expected result – if the idea is that mTOR RNAi does not extend lifespan in fog-2 animals because they have low mTOR pathway activity, how come fog-2 animals are not long lived (as reported by Kenyon lab)? More rigor and discussion are needed here, and if the authors want to make claims about the functional involvement of the mTOR pathway, the authors should analyze at least one more mTOR pathway mutant, e.g., raga-1.

Thank you for this suggestion. Both our microarray and RNA-seq data showed that *fog-2* unmated worms express higher levels of most mTOR components, which is why we initially only performed *let-363* RNAi in the *fog-2* background, but we have now included the lifespan of *let-363* RNAi on N2 worms after brief mating (Figure 5—figure supplement 1 A-B, Supplementary file 1 – experiment 32-33).

We have now also tested several other components of mTOR complex, including *rict-1, daf-15*, and *raga-1*, and found that in all these mutants, reducing the expression of mTOR components in the *fog-2* background protects the hermaphrodites from brief mating induced death (Figure 5E-F, Figure 5—figure supplement 1C-E), supporting our original findings (more replicates are ongoing, but for the purposes of showing the reviewer what progress we have made, these single replicates are currently in the supplement).